# Photothermal effects control ultrafast charge transport in titanium carbide MXenes

Wenhao Zheng [1,2,3], Hugh Ramsden [4], Stefano Ippolito [5], Max van Hemert [4], Danzhen Zhang [5], Teng Zhang[5], Dongqi Li [6], Guanzhao Wen[1], Jaco J. Geuchies [1,7], Minghao Yu [6,8], Xinliang Feng [6,8], Yury Gogotsi [5], Klaas-Jan Tielrooij [4,9] & Hai I. Wang [1,10] ✉

Titanium carbide MXene ($Ti_3C_2T_x$) is an emerging metallic material with promise for (opto)electronics and thermal management. Yet how photoexcitation −particularly via photogenerated thermal energy−modifies its charge carrier dynamics remains poorly understood. By combining time-resolved terahertz spectroscopy and transient reflectance measurements, we reveal a long-lived, photo-induced suppression of conductivity, which we attribute to efficient lattice heating and slow heat dissipation in $Ti_3C_2T_x$. A systematic variation of pump photon energy reveals that this 'negative' photoconductivity can equivalently be induced by lattice temperature increases, indicating a thermal origin. Repetition-rate-dependent transient reflectance measurements further show residual heat persisting over 100 ns, substantially longer than in conventional metals. Our work presents a unified understanding of photothermal effects in $Ti_3C_2T_x$ and their influence on non-equilibrium charge transport, underscoring its potential for photothermal electronics and light-to-thermal energy storage applications.

In conventional single-element metals, *e.g.*, gold or silver, following optical excitation, the initial non-thermalised energetic charge carriers rapidly transfer their energy to "cold" carriers near the Fermi surface, leading to a thermalised Fermi-Dirac distribution on a ~100 femtoseconds (fs) time scale[1–4]. The resulting hot carriers are susceptible to enhanced charge scattering by, *e.g.*, phonons or defects, which transiently reduces their carrier mobility ($\mu$) and thus the conductivity ($\sigma$), as described by $\sigma = e \cdot N \cdot \mu$, where $e$ is the elementary charge and $N$ is the carrier density. This photo-induced conductivity suppression, commonly referred to as negative photoconductivity (NPC; see Fig. 1, process I to II), has been widely observed in metals and doped graphene[5–7], with lifetimes ranging from a few picoseconds (ps) in graphene to hundreds of ps in metals[2,4,8–15]. The NPC timescale is closely linked to electronic dynamics, which are coupled to phonon processes: following thermalisation, hot carriers cool and reach thermal equilibrium with the lattice via phonon emission within a few ps. The resulting lattice heat then dissipates over much longer timescales, eventually equilibrating with the substrate or surrounding environment (process II to III and onwards). This well-established photogenerated excitation or energy relaxation pathways provides a foundational benchmark for interpreting more complex photoresponse in emerging low-dimensional metallic systems.

[1]Max Planck Institute for Polymer Research, Ackermannweg 10, Mainz, Germany. [2]Department of Physics, Massachusetts Institute of Technology, Cambridge, MA, USA. [3]GBA Branch of Aerospace Information Research Institute, Chinese Academy of Sciences, Guangzhou, China. [4]Department of Applied Physics, TU Eindhoven 5612, AZ, Den Dolech 2, Eindhoven, the Netherlands. [5]A. J. Drexel Nanomaterials Institute, Department of Materials Science and Engineering, Drexel University, 3141 Chestnut St, Philadelphia, PA, USA. [6]Center for Advancing Electronics Dresden (CFAED) & Faculty of Chemistry and Food Chemistry, Technische Universität Dresden, Mommsenstrasse 4, Dresden, Germany. [7]Leiden Institute of Chemistry, Leiden University, Einsteinweg 55, Leiden, the Netherlands. [8]Max Planck Institute of Microstructure Physics, Weinberg 2, Halle, Germany. [9]Catalan Institute of Nanoscience and Nanotechnology (ICN2), BIST and CSIC, Campus UAB, Bellaterra (Barcelona) 08193, Spain. [10]Nanophotonics, Debye Institute for Nanomaterials Science, Utrecht University, Utrecht, the Netherlands. ✉e-mail: h.wang5@uu.nl

The recent rise of two-dimensional (2D) materials has opened opportunities for studying exotic optoelectronic and phononic excitations down to the monolayer limit. While extensive progress has been made in exploring many-body effects in semiconducting 2D materials under light excitation, considerably less is known about the optoelectronic and photothermal responses of metallic 2D materials, which show promise as atomically thin electrodes and active layers across the broadband electromagnetic spectrum (from the microwave to UV regime)[16–18]. Among 2D metallic layers, transition metal carbides, nitrides and carbonitrides, known as MXenes, have recently garnered substantial interest as promising candidates for optoelectronic applications due to their exceptional charge mobility, tuneable physico-chemical properties (via surface terminations), and strong light absorption characteristics[19–25]. In particular, $Ti_3C_2T_x$ (where $T_x$ is the surface terminations), one of the most studied MXene systems, has received notable attention for its metallic transport properties and intriguing photoresponse behaviour[20,21,26–32]. Following photoexcitation, the conductivity exhibits a sub-picosecond drop; however, this suppressed conductivity persists for an unusually long time, extending well beyond the nanosecond (ns) timescale, far longer than typical metallic systems. This anomalously long-lived negative photo-conductivity, reported by Li et al. and corroborated by our recent findings[26,27], suggests the presence of unconventional energy retention mechanisms in $Ti_3C_2T_x$. While the observed NPC is consistent with its metallic nature, the anomalously long relaxation lifetime remains poorly understood and leaves open questions. In particular: does the effect originate from long-lived non-equilibrium electronic excitations, *e.g.* hot carriers with slow cooling; or unconventional photothermal properties in MXenes, *e.g.*, inefficient heat dissipation? Employing different ultrafast spectroscopies, including transient absorption (TA) spectroscopy[33–36], time-resolved electron and X-ray diffraction[37], recent studies have suggested slow thermal energy dissipation in $Ti_3C_2T_x$ far beyond the ns timescale.

While several prior studies have independently explored the electrical and thermal properties of MXenes, a cohesive physical picture—one that elucidates how thermal effects influence, and are coupled with, electrical conductivity—remains elusive. Specifically, it is unclear whether (and how) the reported slow thermal energy dissipation underlies the long-lived negative conductivity modulation observed in these materials. To resolve this, an ultrafast investigation that complementarily tracks both charge carrier dynamics and thermal relaxation following photoexcitation is essential.

Here, we present a comprehensive study of photoconductivity dynamics (sub-ps to ns) together with thermal relaxation (ps to 100s of ns), by uniquely combining ultrafast terahertz spectroscopy and transient reflectance measurements. Employing optical pump-THz probe (OPTP) measurements, we show that the conductivity suppression induced by photoexcitation closely matches that produced by raising the lattice temperature, supporting a thermal origin of the long-lived NPC. Consistent with this picture, photon energy ($h\nu$) and fluence-dependent OPTP measurements reveal that identical long-lived NPC is obtained across pump wavelengths when the total absorbed photoenergy is maintained constant. Our ultrafast conductivity studies pinpoint the efficient photothermal response (Fig. 1, I to II) and slow thermal energy dissipation (Fig. 1, II to III) as the main origins of the observed long-lived NPC effect. Finally, to investigate the thermal energy dissipation process of $Ti_3C_2T_x$, we perform transient reflectance with a tuneable repetition rate and find residual heating from prior pulses persisting over 100 ns. Our finding suggests that heat dissipation is slow, in line with the long-lived negative electrical response observed by THz spectroscopy. This inefficient thermal dissipation renders $Ti_3C_2T_x$ as an effective phonon heat reservoir, where absorbed light energy is retained within the material as phononic heat. Our unified framework for coupled thermal and electrical properties in MXenes unlocks opportunities in thermoelectrics, photothermal catalysis, and infrared/terahertz sensing, where synergistic control of heat and charge transport is critical.

## Results and discussion

We first investigate the equilibrium electronic properties through the temperature ($T$)-dependent conductivity measurement using terahertz time-domain spectroscopy (THz-TDS), performed without photo-excitation. This bare thermo-electrical characterisation of how heat affects charge transport properties offers valuable insights into how the carrier mobility evolves with temperature. We use a spray-coated $Ti_3C_2T_x$ thin film with a thickness of ~25 nm, measured by atomic force microscopy, as a model system (see "**Methods**"). In the THz-TDS study, we map out and compare the amplitude and phase of the THz electric field transmitted through the pristine substrate alone ($E_{ref}$) and the sample on the substrate ($E_{sample}$) across various temperatures (as shown in Fig. 2a). The thin $Ti_3C_2T_x$ films exhibit ~18.5% absorption across 0.5–2.2 THz, consistent with the thin-film electromagnetic limit (see Supplementary Fig. 1)[30]. As the temperature decreases, the amplitude of the transmitted THz pulses further diminishes, indicating increased absorption by free carriers. This result suggests that the conductivity of $Ti_3C_2T_x$ increases upon lowering the temperature. To enable a quantitative analysis, we convert the time-domain data into frequency-resolved complex-valued conductivity by performing a Fourier transform under the thin-film approximation (see **Methods**). We then analyse the complex conductivity by applying the Drude model (see Fig. 2b). This analysis yields the scattering time ($\tau$) and plasma frequency ($\omega_P$) at various temperatures, as summarised in Fig. 2c and Supplementary Fig. 2, respectively. The charge carrier scattering time decreases with $T$, indicative of band-like charge transport[27]. Furthermore, we observe a direct one-on-one correlation between the scattering time and the conductivity amplitude, which suggests that changes in conductivity ($\sigma = e \cdot N \cdot \mu \propto N \cdot \tau$) with temperature are predominantly governed by variations in the scattering time $\tau$, and consequently the mobility $\mu$ (since $\mu \propto \tau$). In this regard, we find that the inferred $\omega_P$ is unchanged in the same temperature regime as shown in Supplementary Fig. 2: since the carrier population $N$ is proportional to $\omega_P^2$, this result indicates that $N$ remains nearly constant in the measured $T$ range. Additionally, using an effective carrier mass $m^*$ of 0.28 $m_o$,[38] we estimate the charge carrier mobility and intrinsic carrier density in $Ti_3C_2T_x$ to reach ~70 cm$^2$V$^{-1}$ s$^{-1}$ and 1.2 × 10$^{21}$ cm$^{-3}$ at room temperature, respectively. These values are comparable to previous results obtained by THz spectroscopy (~34 cm$^2$V$^{-1}$ s$^{-1}$ and 2 × 10$^{21}$ cm$^{-3}$)[39].

The pronounced temperature dependence of the charge carrier dynamics reveals that electron−phonon scattering likely dominates the transport properties in $Ti_3C_2T_x$. As the temperature decreases, the reduced phonon populations lead to a substantial suppression of the scattering rates, thereby enhancing the carrier mobility and conductivity. This behaviour suggests the central role of electron−phonon interactions in governing the electrical response of the system. Using the scattering times obtained from the Drude analysis, we estimate the in-plane mean free path as $l(T) = v_F \tau(T)$, where $v_F = \frac{\hbar k_F}{m}$ and $k_F = (3\pi^2 n)^{1/3})$. Here $\hbar$, $k_F$, $v_F$ are the reduced Planck constant, Fermi wavevector and Fermi velocity. Based on these parameters, the mean free path ranges from ~14 nm at 300 K to ~23 nm at 100 K.

After examining the equilibrium transport behaviour, we turn to investigate the transient photoexcited carrier response, *i.e.*, optoelectronic properties employing ultrafast OPTP spectroscopy. In particular, by varying the pump-photon wavelength and fluence, we generate hot carriers having a prescribed initial energy and track the subsequent energy relaxation dynamics in $Ti_3C_2T_x$ MXene thin films. In OPTP measurements, an optical pump pulse with tuneable photon wavelength (319 to 1750 nm or photon energy $h\nu$ ranging from 0.7 to 3.88 eV) excites charge carriers in the material. A single-cycle THz pulse subsequently probes the time-resolved photoconductive response ($\Delta\sigma = \sigma_{pump} - \sigma_0$, where $\sigma_0$ and $\sigma_{pump}$ are conductivities

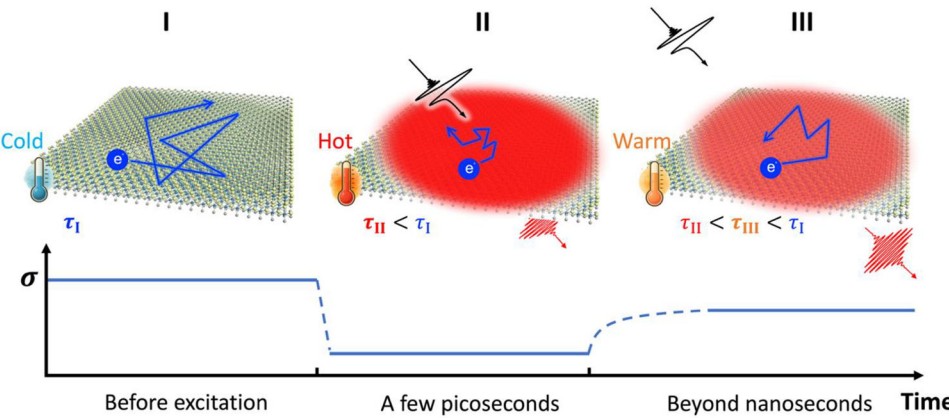

**Fig. 1 | Non-equilibrium photoexcitation dynamics.** Schematic illustration of the photoconductivity evolution in a Ti$_3$C$_2$T$_x$ MXene following optical excitation. **I.** Before photoexcitation, the system exhibits relatively long carrier scattering time ($\tau$) and mean free path, and thus relatively high conductivity. **II.** Within a few ps after photoexcitation, efficient light-induced lattice heating takes place. This results in a suppression of the electrical conductivity, corresponding to reduced scattering time and mean free path. **III.** The conductivity remains suppressed even beyond nanoseconds, due to the slow thermal energy relaxation. The corresponding time evolution of conductivity ($\sigma$) is shown below each time scale.

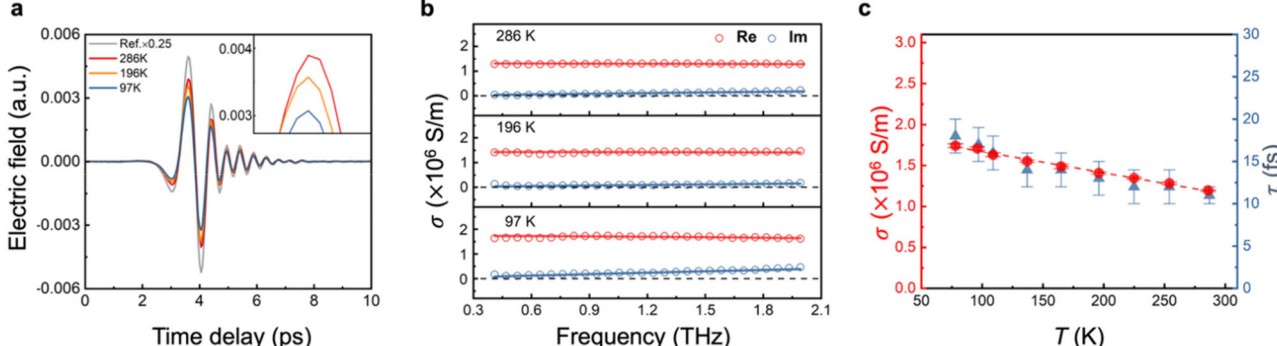

**Fig. 2 | Temperature-dependent static conductivity in Ti$_3$C$_2$T$_x$.** **a** Time-domain static terahertz signals for the bare substrate (as reference) and the MXene thin film, measured at various temperatures under vacuum conditions (pressure $< 1.8 \times 10^{-4}$ mbar). **b** Frequency-dependent complex conductivity spectra extracted via Fourier transform of the data in panel a, using the thin-film approximation. Red and blue circles denote the real and imaginary components of the conductivity, respectively. **c** Temperature dependence of the conductivity $\sigma(T)$ (red data, left axis) and charge carrier scattering time $\tau(T)$ (blue dots, right axis), extracted from Drude model fits to the complex conductivity spectra. Error bars represent one standard deviation uncertainties of the fitted parameters obtained from the Drude model fitting. The red line indicates a linear fit to $\sigma(T)$, highlighting its monotonic decrease with increasing temperature, likely due to enhanced electron–phonon scattering.

before/after photoexcitation) of charge carriers at varying pump-probe delay times. The photoconductivity is monitored by characterising the differential transmission of the THz signal, following the thin-film approximation (**Methods** for details). Figure 3a illustrates the $h\nu$-dependent photoconductivity dynamics in Ti$_3$C$_2$T$_x$ MXene thin films with the same absorbed photon density ($N_{abs}$). All measurements are conducted in the low pump fluence regime ($< 120~\mu J/cm^2$), ensuring that the photoconductivity increases linearly with fluence at each wavelength. We observe a rapid, sub-picosecond initial decay component in the photoconductivity dynamics. As shown in Supplementary Fig. 3, the spectral weight of this fast photoconductivity response, obtained by subtracting the average value between 2-5 ps from the peak photoconductivity magnitude, exhibits a super-linear dependence on photon energy at a fixed absorbed photon density. We attribute this fast initial decay to the ultrafast energy relaxation between thermalized/nonthermalized hot carriers to the lattice, resulting in ultrafast heating of the lattice, which is in line with recent ultrafast studies by TA and X-ray scattering studies[35,37,40]. Although different excitation schemes (e.g., interband vs. plasmonic) may initiate distinct energy relaxation pathways, all the energy ultimately

deposits into a common phonon bath within this 2–3 ps window (corresponding to the process from stage I to II in Fig. 1). The later conductivity recovery (stage II → III) occurs on the longer timescales, corresponding to slow lattice heat dissipation. This result indicates that a strong electron-phonon coupling dominates the sub-ps response in MXenes, which leads to an efficient conversion of photo-excited energy into the phonon system, thus facilitating efficient lattice heating in MXenes. Beyond this initial relaxation phase, the system enters a distinct long-lived regime: following the peak photoconductivity, the dynamics exhibits a relatively slow decay that persists up to the ns timescale. After the ultrafast carrier–phonon equilibration (sub-picosecond after photoexcitation), the remaining photoconductivity scales linearly with photon energy $h\nu$, as shown in Fig. 3b for the data averaged between a 3-10 ps window. This linear dependence persists at later times as well, including in the 200–1000 ps range (see Supplementary Fig. 4), indicating a consistent correlation between excitation energy and photoconductivity amplitude across different timescales. By combining the linear scaling between the long-lived photoconductivity and $h\nu$ (at a given $N_{abs}$) and the observation that the long-lived photoconductivity itself also

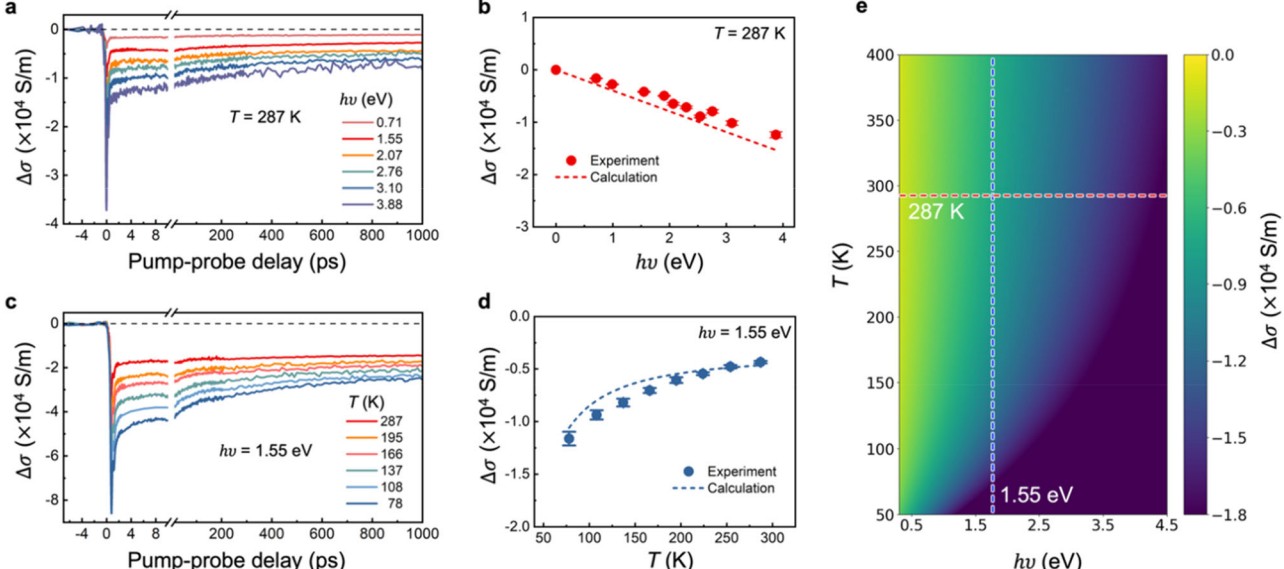

**Fig. 3 | v- and temperature-dependent photoconductivity in titanium carbide MXene. a** $h\upsilon$-dependent photoconductivity dynamics in $Ti_3C_2T_x$. **b** The averaged photoconductivity (between 3 to 10 ps) as a function of $h\upsilon$. The dashed line is from the calculation result presented as the vertical cut in panel (**e**). **c** Temperature-dependent photoconductivity dynamics in titanium carbide. **d** The averaged photoconductivity (between 3 to 10 ps) as a function of temperature. The dashed line is the horizontal cut in panel e. Error bars in panels b and d represent the standard deviation of the photoconductivity values averaged over the time window from 3 to 10 ps. **e** Calculated photoconductivity as a function of temperature and $h\upsilon$, with horizontal and vertical dashed cuts shown in b and d to compare with the experimental data.

increases linearly with $N_{abs}$, this result indicates that the same transient conductivity states can be achieved using different pump wavelengths, provided that the absorbed photon power remains constant. As shown in Supplementary Fig. 5, we demonstrate this statement by comparing the same long-lived photoconductivity by pumping 1.55 vs 3.1 eV with the same absorbed pump power. These results provide compelling evidence that the prolonged photothermal effect, where light excites electrons that quickly decay by transferring their excess energy into lattice heat, likely underlies the observed sustained negative photoconductivity.

To further substantiate this photothermal interpretation, we perform temperature-dependent photoconductivity studies using OPTP measurements on $Ti_3C_2T_x$ MXene thin films, keeping the pump photon energy and fluence fixed. As shown in Fig. 3c, d the amplitude of the long-lived photoconductivity ($\Delta\sigma$) becomes increasingly negative as the temperature decreases from 287 to 78 K. This temperature dependence can be rationalised by the reduced lattice heat capacity at lower temperatures[37,41]. For a fixed amount of absorbed energy, a lower thermal capacitance leads to a larger temperature rise ($\Delta T$), thereby enhancing the photothermal effect.

We further turn to a unified model (see the logic flow in Suppl. Fig. 6) that includes the impact of photoexcitation and microscopic transport parameters (*e.g.*, lattice temperature and scattering time) in determining carrier conductivity, enabling a direct comparison with the photoconductivity data presented in Fig. 3a–d. The key idea behind our model relies on the knowledge of the initial microscopic transport parameters ($N$, $\mu(T)$, based on Fig. 2), which dictate the charge conductivity ($\sigma = \frac{Ne^2\tau}{m^*}$). Upon light absorption, which then leads to lattice heating, or upon direct lattice temperature variation, we compute the energy injected into the system. Combining this with the material's thermal capacity, we then derive the resulting temperature changes. We can then calculate the photon energy $h\upsilon$ and temperature-dependent photoconductivity following:

$$\Delta\sigma = \sigma_{pump} - \sigma_0 = \sigma(T_0 + \Delta T(h\upsilon, T)) - \sigma(T_0) \qquad (1)$$

where $\sigma(T_0)$ is the $T$-dependent static conductivity (from Fig. 2c), and $\Delta T(h\upsilon, T)$ is the lattice temperature rise resulting from energy transfer from absorbed photons to electrons and subsequently to the lattice. As $N$ is $T$-independent, the primary change in $\sigma$ originates from the pump or $T$-induced scattering time change $\Delta\tau$. Here we assume 100% energy transfer efficiency, meaning that all absorbed energy from photoexcitation is eventually transferred to the lattice after the rapid (~ps) carrier–lattice thermalisation. The temperature rise $\Delta T(h\upsilon, T)$ is calculated from the pump energy (related to $N_{abs} \times h\upsilon$) and the known $T$-dependent thermal capacity of $Ti_3C_2T_x$ MXenes from previous reports[42–44].

The calculation results, performed without any free-fitting parameters, show good agreement with the experimental results, as shown in Fig. 3b and d: in Fig. 3e, we present the simulated $\Delta\sigma(h\upsilon, T)$, with vertical and horizontal cuts matching the photon energy and temperature–dependent photoconductivity observed in Fig. 3b, d, respectively. Together, our combined pump-fluence ($N_{abs} \times h\upsilon$) and temperature-dependent photoconductivity studies consistently demonstrate that the long-lived negative photoconductivity is fundamentally associated with the initial energy deposited into the system via photoexcitation (~ $N_{abs} \times h\upsilon$, by controlling fluence or pump wavelengths). It is governed by the intrinsic thermal and electronic properties of MXenes—namely, their temperature-dependent heat capacity and charge scattering time. Once these temperature-dependent microscopic electronic and thermal properties of MXenes are known, we can predict and control the conductivity dynamics by tailoring the initial energy deposited through photoexcitation. This insight highlights the central role of ultrafast electron–phonon coupling and slow thermal dissipation in shaping the optoelectronic response of metallic 2D systems.

In conventional metals, ultrafast optical studies typically reveal a multistep relaxation: sub-ps electron–electron thermalisation, followed by sub-ps to ps electron–phonon energy transfer, and finally ps-to-hundreds-of-ps lattice cooling. While this cooling in ultrathin films (≤ ~10 nm) is often reported to be shortly lived from few to 100 s of ps depending on the metal nature and its interfaces[12,15], our ~ 25 nm $Ti_3C_2T_x$ films exhibit a drastically slow dynamics persisting for ns

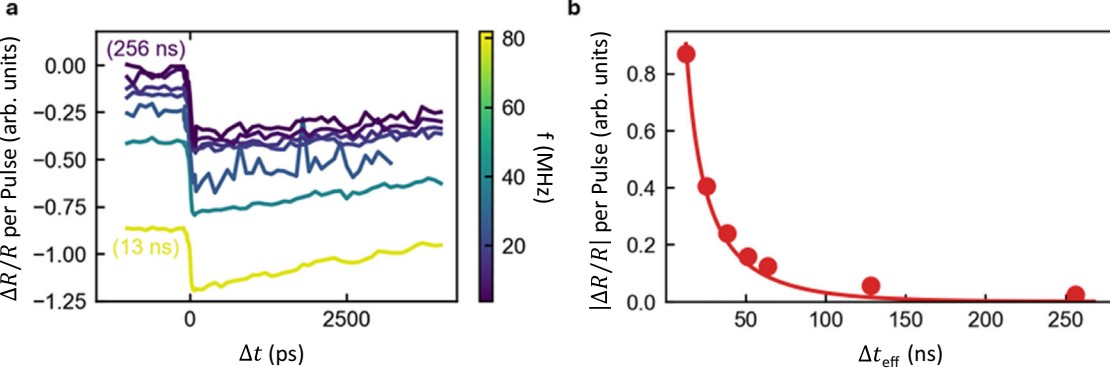

**Fig. 4 | Transient reflectivity measurement of heat dissipation. a** Influence of repetition rate on transient reflectivity. At $\Delta t = 0$ there is a negative step in $\Delta R/R$ as heat is introduced into the system by the pump. For $\Delta t < 0$ we examine the influence of the previous pump pulses (see S7b for details). For the highest repetition rate, a significant residual difference in reflectivity is seen from the previous pump. For longer time delays, the influence of the previous pump reduces. Effective delay time ($\Delta t_{eff}$) for the highest (13 ns) and lowest (256 ns) repetition rates are also indicated. **b** Residual $|\Delta R/R|$, at long effective delays – extracted by averaging $|\Delta R/R|$ between a delay of $-1$ and $-0.1$ ns. For effective delays over 100 ns, we still measure some influence from the previous pump pulses, indicating some heat from the pump is still present. We perform a fit, plotted with the solid red line, using a simple model that accounts for the accumulation of heat from previous pump pulses. See Methods for details. With this, we extract a decay time of $40 \pm 1$ ns.

timescale. This orders-of-magnitude difference indicates a severe thermal bottleneck, which may be attributed to low interfacial conductance and/or low in-plane thermal conductivity in our MXenes[35,37,45] and a very low emissivity of $Ti_3C_2T_x$[46].

To confirm the thermal origin of the photoconductivity dynamics observed in Fig. 3, we investigate the opto-thermal properties using transient reflectivity measurements (TRMs). While OPTP captures dynamics only up to ~1 ns, in our measurement scheme, transient reflectivity allows us to access long-lived thermal dynamics, up to hundreds of ns. We make use of an optical pump at 600 nm, and an optical probe at 790 nm, where the reflectivity is dominated by a plasmon resonance[47]. This response has been reported to be temperature dependent[48], thus, the change in reflectivity induced by the pump $\Delta R/R$ indicates light-induced temperature changes in the sample (see Supplementary Fig. 8).

In Fig. 4a, we show $\Delta R/R$ as a function of pump-probe delay time, $\Delta t$, using a range of pulse repetition rates. For all repetition rates, at $\Delta t = 0$, we see a sharp negative step in $\Delta R/R$. We attribute this to rapid heating of the lattice caused by the pump. This is consistent with observations in Fig. 3, where conversion of the optical pump to thermal energy in the lattice is shown to proceed on a sub-picosecond timescale. We cannot resolve this in our transient reflection measurements, which have a time resolution of ~30 ps. The heat slowly decays as it spreads in both in- and out-of-plane directions[45,48]. For $\Delta t < 0$, we see that for shorter times between subsequent pump and probe pulses, meaning higher repetition rates, $\Delta R/R$ does not return to zero, indicating that heat from previous pump pulses is still present.

As described in detail in Supplementary Fig. 7b, measurements at a small negative time delay are equivalent to an effective time delay $\Delta t_{eff}$ that is equal to the time between subsequent pump and probe pulses. With this insight, in Fig. 4b, we plot the residual signal across various $\Delta t_{eff}$ values, controlled via the variable repetition rate. The decay of this residual signal remains significant for effective time delays over 100 ns. To quantify this decay, we use a simple model, described in the Methods, which accounts for the accumulation of heat from successive pulses. This model matches well with our data and gives a decay constant of $40 \pm 1$ ns. This slow decay is indicative of inefficient diffusion of heat into the substrate, likely due to high thermal boundary interfacial conductance. Our value is consistent with the previous reports of Wang et al.[35], who similarly report a decay constant around 50 ns for 25 nm thick $Ti_3C_2T_x$.

The relatively slow heat dissipation observed from near time zero to hundreds of ns suggests that the dominant energy carrier is phononic heat, as electronic heat would dissipate much more rapidly. This supports the OPTP findings, which also directly point to a phononic character of the thermal energy. Given the thin nature of our multi-layered MXene (with thicknesses of 25 nm and lateral size of several μm), the >100 ns slow heat dissipation implies a thermal transport bottleneck in our samples, at least in the out-of-plane direction. This effect may originate from (i) a small interfacial (Kapitza) conductance across $Ti_3C_2T_x$/substrate interfaces[37], and/or (ii) a low inter-layer thermal conductivity between flakes[35]. A quantitative separation of interfacial and cross-plane thermal conductance will require dedicated thermal metrology (e.g., TDTR/FDTR) studies with systematic thickness/substrate series, which we plan to address in a follow-up study. Consequently, the MXene acts as an efficient phonon heat reservoir: optical energy is swiftly converted into lattice heat, which remains trapped within the material, presumably due to poor thermal coupling across the interface.

In summary, we employed pump wavelength- and temperature-dependent OPTP spectroscopy to investigate the dynamics of photoexcited carriers in $Ti_3C_2T_x$ MXene, emphasising the energy relaxation pathways. Upon light absorption, efficient conversion of optical energy into heat leads to ultrafast lattice heating. The elevated lattice temperature enhances carrier–phonon scattering, thereby impeding charge transport and producing negative photoconductivity. In $Ti_3C_2T_x$, the unusually slow thermal energy dissipation prolongs this elevated-temperature state, resulting in the observed long-lived photoconductivity suppression. We further support this model quantitatively by tuning the initial photoexcitation energy and exploiting the temperature-dependent electronic and thermal properties of MXenes to achieve full control over the photothermal-electric effects in $Ti_3C_2T_x$. Furthermore, through transient reflectivity measurements, we demonstrate this heat dissipation is remarkably slow, with heat signatures being detected over 100 ns after initial excitation.

Our findings reveal the impact of slow thermal relaxation on its electrical effects in $Ti_3C_2T_x$ MXene, providing insights into the fundamental photophysical processes relevant to MXene-based photothermal electronics. For instance, the observed ultra-slow thermal energy dissipation and high electrical conductance render MXenes relevant for thermoelectrics or photothermal catalysis. In addition, the strong temperature-dependent momentum scattering time and the

slow heat dissipation in MXenes make them also promising for heat, far-IR or THz sensing applications, potentially even in the *dc* limit given the slow pump-induced dynamics. Our combined electrical and thermal transport studies not only unveil the fundamental photophysics in MXenes, but also highlight opportunities for using them in various optoelectronics or photothermal applications.

## Methods

### Material preparation

1 gram of stoichiometric $Ti_3AlC_2$ MAX powder (produced following our previously reported synthesis protocols)[49] was slowly added to a 20 mL mixture of concentrated HF (49 wt.%), concentrated HCl (36 wt.%), and deionized water, with a volumetric ratio of 1:6:3. The MAX phase was etched at 35 °C for 48 h. After the reaction was completed, the etching product was washed with deionized water via multiple centrifugation steps (3500 rpm, 5 min) until neutral pH. To delaminate the MXene sheets, the $Ti_3C_2T_x$ multilayer product was stirred at 300 rpm in a 50 mL aqueous solution containing 1 gram of LiCl at 35 °C for 18 h. Then, LiCl was removed from the solution by a single centrifugation step (3500 rpm, 15 min), discarding the clear supernatant. After that, $Ti_3C_2T_x$ was redispersed in water via manual shaking and centrifuged multiple times for 5 min at 3500 rpm (to precipitate the multilayer MXene and unreacted MAX particles). For each step, the dark supernatant (containing single-layer MXene sheets) is collected.

### Material deposition

Before using, the substrates are cleaned by sonication in acetone and 2-propanol for 10 min each. Then, they are heated on a hot plate at 350 K. $Ti_3C_2T_x$ aqueous dispersions (concentration $\approx 0.5$ mg/mL) are sprayed by using a commercial airbrush gun, with a needle and nozzle diameter equal to 0.3 mm. The distance between the nozzle tip and the substrate is kept $\approx 20$ cm[22,49].

### Terahertz time-domain spectroscopy (THz-TDS)

We performed the terahertz measurements by a femtosecond laser system, which produces 50-fs pulses at 800 nm with a 1 kHz repetition rate. For terahertz generation, we used optical rectification in a ZnTe crystal oriented along the <110> direction, with detection handled by electro-optic sampling in a second ZnTe crystal.

For our experiments, the samples were placed in a nitrogen-purged environment to avoid absorption from water vapour. The THz pulses were detected in transmission mode, and the amplitude and phase of the terahertz electric field were recorded. The frequency-dependent conductivity spectra were derived using the Fourier transformation of the time-domain signals. All measurements were performed at room temperature unless otherwise specified. The refractive index of the substrate was measured to ensure accurate extraction of the terahertz response.

To enable a quantitative analysis, we convert the time-domain data into frequency-resolved complex-valued conductivity by performing a Fourier transform under the thin-film approximation, as described in the followed equation:

$$\sigma(\omega) = \frac{1 + n_s}{Z_0 d} \left( \frac{E_{ref}(\omega)}{E_{sample}(\omega)} - 1 \right) \tag{2}$$

where $n_s$ is the refractive index of the substrate in the THz range, $d$ is the sample thickness, and $Z_0$ is the impedance of free space, definied as $Z_0 = (\varepsilon_0 c)^{-1} \approx 377\Omega$.

To further analyse the intrinsic charge transport behaviour, we fit the extracted complex conductivity spectra using the Drude model expressed in terms of the plasma frequency and scattering time:

$$\sigma(\omega) = \frac{\omega_P^2 \varepsilon_0 \tau}{1 - i\omega\tau} \tag{3}$$

where $\varepsilon_0$ is the vacuum permittivity, $\omega_P$ is the plasma frequency, and $\omega$ is the angular frequency of the terahertz field. The plasma frequency is related to the free carrier density $N$ by: $\omega_P^2 = \frac{Ne^2}{\varepsilon_0 m^*}$

where $e$ is the elementary charge and $m^*$ is the effective mass of the charge carriers. By fitting both the real and imaginary parts of the conductivity $\sigma(\omega)$ simultaneously, we extract the key transport parameters, including $\omega_P$ and $\tau$, which reflect the carrier density and scattering rate, respectively.

The effective carrier mass $m^* = 0.28\, m_0$ is used, which obtained from the DFT-calculated value for $Ti_3C_2(OH)_2$[49]. We note that experimentally synthesised $Ti_3C_2T_x$ contains mixed surface terminations (−O, −OH, −F, −Cl), which may alter the band dispersion and therefore $m^*$.

### Optical pump-terahertz probe spectroscopy (OPTP)

OPTP spectroscopy was performed using a Travelling-wave Optical Parametric Amplifier of Superfluorescence (TOPAS), driven by a Ti amplifier operating at 800 nm, with a pulse duration of ~100 fs as the pump beam. The TOPAS system generates tuneable pump pulses in the visible to infrared range, which were synchronised with the THz probe pulses for the measurements. The pump-induced changes in the terahertz transmission were monitored by varying the delay between the pump and probe pulses using a mechanical delay stage. The resulting photoconductivity was calculated using the thin-film approximation. The entire measurement setup was kept under a nitrogen atmosphere or placed under vacuum to avoid water vapour absorption in the terahertz range.

The photoconductivity can be monitored by characterising the differential transmission of the THz signal, following:

$$\Delta\sigma = -\frac{1 + n_s}{Z_0 d} \cdot \frac{E_{pump} - E_0}{E_0} \tag{4}$$

where $E_{pump}$ and $E_0$ represent the peak intensity of the transmitted THz field with and without photoexcitation.

To make a fair comparison for the photoconductivity, we normalised photoconductivity dynamics to the absorbed photon density $N_{abs}$. The wavelength-dependent absorption is obtained by:

$$N_{abs}(h\upsilon) = \frac{P_{avg}/f}{h\upsilon} \times A(h\upsilon) \div A_{spot} \tag{5}$$

where $P_{avg}$ is the average laser power, $f$ the laser repetition rate (500 Hz) after optical chopper, and $A_{spot}$ the pump spot area overlapped with our THz probe, which we determined by using a 1 mm diameter calibrated iris aperture, $A(h\upsilon)$ is the film absorption measured by UV–Vis−NIR spectrophotometry with an integrating sphere. Before this, we checked the fluence dependence at each wavelength and selected a fluence range where the OPTP signal is linear.

### Transient reflectivity measurements (TRM)

Transient reflectivity measurements were taken on a custom setup, schematically illustrated in Supplementary Fig. 7a. Our laser is a SuperK FIANUM FIU 6 supercontinuum laser, with the variable repetition rate option. The pump (600 nm) and probe (790 nm) beams are selected from the white light using the SuperK SELECT AOTF module. These are separated using a dichroic Thorlabs DMLP650 dichroic mirror into separate paths. The pump path goes through a chopper at 4370 kHz, which is synchronized with a Zürich Instruments MFLI lock-in amplifier. It then passes over a Standa 8MT−195X-1040 delay line, before it is recombined with the probe line. Both beams are focused on the sample using an OptoSigma PAL-50-NIR-HR-LC00 objective, which also collects the reflected light. The reflection of the probe is sent, using a beamsplitter, through a 650 nm longpass filter to a Thorlabs

DET10A2 photodiode. The signal from the photodiode is collected using a lock-in amplifier.

## Modelling of accumulated TRM signal

To extract the decay time from the transient reflectivity at long effective delays, we developed a model that accounts for the accumulation from the $N_p$ pump pulses that arrive within the chopper modulation window. Treating each pulse separately, each pump pulse induces a change in reflectivity with amplitude A, which h decays exponentially with a decay time constant $t_{delay}$. After the chopper modulation window is over, the first pulse will have decayed over $N_p/f$ seconds, the second by $(N_p - 1)/f$ seconds, up to the $i$th pulse at $(N - 1)/f$ seconds, where $f$ is the repetition rate of the laser. Thus, up to the final pulse, the accumulated effect can be expressed by:

$$\frac{\Delta R}{R} = \sum_{i=0}^{N_p} A e^{-\frac{(N_p - i)}{f \cdot t_{delay}}} = \sum_{i=0}^{N_p} A e^{-\frac{i}{f \cdot t_{delay}}} \tag{6}$$

To fit the data, we fix A to match the start of the decay, set $N_p$ and $f$ according to the pulse repetition rate and chopper modulation frequency and fit the value of $t_{delay}$.

## Data availability

All data are available in the main text or the supplementary materials. Source data are provided in this paper.

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

## Acknowledgements

W.Z. acknowledges support from the German Research Foundation through a Walter Benjamin fellowship (grant no. Z.H. 1262/1-1). J.J.G. gratefully acknowledges financial support from the Alexander von Humboldt Foundation. G.W. acknowledges the fellowship support from the China Scholarship Council (CSC). H.I.W. acknowledges funding support from Utrecht University. W.Z., J.J.G., G.W., and H.I.W. acknowledge funding support from the Max Planck Society. We gratefully acknowledge Mischa Bonn and all members of the THz group in the Department of Molecular Spectroscopy at MPIP for insightful discussions. We also thank Zihui Lu (GBA Branch of the Aerospace Information Research Institute, Chinese Academy of Sciences) and Ioannis Georgoulis (Utrecht University) for valuable discussions. The ICN2 is funded by the CERCA programme / Generalitat de Catalunya. The ICN2 is supported by the Severo Ochoa Centres of Excellence programme, Grant CEX2021-001214-S, funded by MCIU/AEI/10.13039.501100011033. K.J.T. acknowledges funding from the European Union's Horizon 2020 research and innovation programme under Grant Agreement No. 101125457 (ERC CoG "EQUATE") and Spanish MCIN/AEI project PID2022-142730NB-I00 "HYDROPTO").

## Author contributions

H.I.W. conceived and designed the project. W.Z. conducted the terahertz spectroscopy experiments, analysed the data and performed the modelling under the supervision of H.I.W., with input from K.J.T. and additional assistance from G.W. and J.J.G. M.v.H. and H. R. performed transient reflectivity measurements under the supervision of K.J.T. S.I., D.Z., T.Z. and D.L. synthesised samples under the guide of M.Y., X.F. and Y.G. W.Z., H.R., K.J.T. and H.I.W. wrote the manuscript with input from all other authors.

## Funding

## Competing interests

The authors declare no competing interests.
