## [Transparent Peer Review file · Nature Communications]

Photothermal effects control ultrafast charge transport in titanium carbide MXenes

Corresponding Author: Professor Hai Wang

Version 0:

Reviewer comments:

Reviewer #1

(Remarks to the Author)

This work reports an anomalously long thermal energy relaxation time exceeding 100 ns in metallic Ti₃C₂T_x, which is a remarkable finding with significant scientific implications. The results highlight new opportunities for using such materials in various optoelectronic, photothermal, heat-sensing, far-infrared, and terahertz sensing applications. The study is overall compelling; therefore, I recommend its acceptance after the authors address the following points:

1. The manuscript should clearly distinguish between the mechanisms of “shielding” and “absorption” in the THz response. The claimed absorption of 80% significantly exceeds the theoretical limit (~50%) for intrinsically absorbing films with subwavelength thickness.
2. The effective carrier mass value ($m^*=0.28 m_0$) used in the analysis is derived from theoretical calculations for Ti₃C₂(OH)₂. However, experimentally synthesized Ti₃C₂T_x typically contains a mixture of various functional groups (e.g., -O, -OH, -F, -Cl). It is recommended that the authors perform Hall effect measurements to directly determine the carrier concentration, thereby enabling obtaining of the effective mass, and mobility by THz datas, offering more reliable support for the charge transport properties.
3. The discussion regarding the underlying mechanism for the unusually slow thermal energy dissipation in Ti₃C₂T_x is currently insufficient.
4. The manuscript lacks a direct comparison between the measured thermal energy relaxation time and those typical of conventional metals.

Reviewer #2

(Remarks to the Author)

This manuscript addresses the question of the origin of the anomalously long relaxation lifetime in Ti₃C₂ MXene films. It concludes that the negative photoconductivity arises from thermal effects, consistent with previous studies using time-resolved diffraction and transient absorption. The novelty and significance of this work lie in its comprehensive dataset, which presents photoconductivity measurements across an exceptionally broad range of photon energies, from UV to IR. This contribution is both timely and relevant, offering valuable insights into MXene properties. The manuscript could be further strengthened by expanding the discussion on several points of interest to the community, which would not require additional experiments.

1. In the THz TDS measurements, the conductivity exhibits a Drude-like response, without the low-frequency suppression or negative imaginary conductivity reported in prior studies which used Drude-Smith model to analyze the conductivity. This result suggests a significant improvement in film quality, where flake edges or other defects no longer hinder long-range transport. It would be valuable if the authors could comment on which advances in MXene synthesis or film deposition they attribute to this improvement. A more detailed methods section outlining the steps necessary to achieve such high AC conductivity would also be highly appreciated by the community.
- 2.

The authors further observe that carrier scattering rates decrease substantially as the temperature is lowered, accompanied by a reduction in conductivity. In contrast, other studies have reported that Ti_3C_2 conductivity can increase again at low temperatures (50–100 K) after an initial decrease (see, for example, Anasori et al, *Nanoscale Horiz.* 2016, 1, 227; Figure 3B). Here, the data clearly show no such reversal. Does this difference again highlight film quality as the determining factor? The authors' interpretation would greatly aid the community in deepening the understanding of conductivity mechanisms in these materials.

3. In the OPTP section, the authors present compelling data on the excitation energy dependence of NPC conductivity dynamics. They show that the sub-picosecond component becomes more pronounced at higher excitation photon energies. This dataset substantially extends the previously reported comparison of 400 nm versus 800 nm excitation in Ref. [25]. A question to the authors: in Ref. [25], the rise time of NPC also differed between the two wavelengths. Do you observe a similar effect here? From Fig. 3a, it is difficult to determine. If such an effect is present, is there a clear transition from slower to faster rise times, or is the change monotonic with excitation energy? Any comments on the possible mechanisms driving this behavior would be highly valuable.

4. The central findings and novelty of this work largely stem from the excitation energy-dependent measurements presented in Figure 3. The optical absorption of Ti_3C_2 varies significantly across the 0.5–4.5 eV range explored here, from interband transitions in the UV, to free-carrier absorption in the IR, to the pronounced peak near 750–800 nm attributed to plasmon resonance, which the authors use to extract thermal dynamics from transient reflectivity. A key question is whether the authors accounted for this variation in optical absorption when estimating the initial deposited energy via photoexcitation (N_{abs}). The energy-dependent relaxation reported in Fig. 3b appears linear and does not correlate with the expected absorption spectrum. Why is that? Are there differences in the earlier dynamics (sub-2 ps) that are not captured in the presented analysis? Intuitively, one might expect that exciting interband transitions versus resonant plasmon excitation would yield distinct relaxation pathways at early times. Since the authors already have the relevant data, a discussion of this point, ideally supported by a UV–Vis spectrum, would provide valuable additional insight.

Small comment:

In the legend of Fig. 3, please specify the excitation energy for panel C (and vertical cut in Panel e)

Reviewer #3

(Remarks to the Author)

Zheng et al. conducted a systematic investigation of the photoexcitation dynamics in $\text{Ti}_3\text{C}_2\text{T}_x$ MXenes, employing a combination of optical pump–terahertz probe (OPTP) spectroscopy and transient reflectance (TR) measurements. Their study revealed that photothermal effects play a dominant role in the unusually long-lived negative photoconductivity (NPC) observed in MXene. The main innovation of the work lies in proposing a unified framework that connects photothermal conversion, electron–phonon interactions, and thermal relaxation with charge transport processes, supported by comprehensive experimental data. This work provided significant insights into the photophysical mechanisms of two-dimensional metallic MXenes and offered valuable guidance for their potential optoelectronic and thermoelectric applications. The analysis is mostly convincing, and thus the work is in principle well suited for publication in *Nature Communication*. Nevertheless, the manuscript may have several weaknesses and requires additional description and analysis to strengthen its conclusions.

Q1. The current version of the manuscript lacks sufficient structural and morphological characterization of the $\text{Ti}_3\text{C}_2\text{T}_x$ MXene samples. For example, techniques such as X-ray diffraction (XRD) or transmission electron microscopy (TEM) are needed to verify the sample quality and crystallinity. Without these data, it may be difficult to fully assess the reliability and reproducibility of the observed photoexcitation dynamics.

Q2. As illustrated in Fig. 1, the authors suggested that increasing the temperature reduces the mean free path of charge carriers, which is a reasonable description. In this study, based on the available data, is it possible to estimate the mean free path values of MXene charge carriers in the three stages shown in Fig. 1?

Q3. In the manuscript, the authors state: “Our ultrafast conductivity studies pinpoint the efficient photothermal response (Fig. 1, I to II) and slow thermal energy dissipation (Fig. 1, II to III) as the main origins of the observed long-lived NPC effect.” Could more accurate time scales for these two processes be extracted through quantitative fitting of the dynamical curves?

Q4. The authors state: Fig. 3a illustrates the $h\nu$ -dependent photoconductivity dynamics in $\text{Ti}_3\text{C}_2\text{T}_x$ MXene thin films with the same absorbed photon density (N_{abs}). It would be helpful if the paper could provide a clear definition of absorbed photon density as well as the calculation method used to ensure the same absorbed photon density under different excitation wavelengths. Since MXenes exhibit varying absorption strengths across different spectral regions, it is important to explain how the authors corrected for these differences to guarantee consistent absorbed photon density across all wavelengths.

Version 1:

Reviewer comments:

Reviewer #1

(Remarks to the Author)

Authors have addressed the issues I raised previously. I would like to recommend publication of this work.

Reviewer #2

(Remarks to the Author)

Authors have addressed my concerns and questions. I support publication.

Reviewer #3

(Remarks to the Author)

The authors addressed all questions raised by reviewers, and revised accordingly in the main text. I recommend publishing as it.

Reviewer #1 (Remarks to the Author):

This work reports an anomalously long thermal energy relaxation time exceeding 100 ns in metallic $\text{Ti}_3\text{C}_2\text{T}_x$, which is a remarkable finding with significant scientific implications. The results highlight new opportunities for using such materials in various optoelectronic, photothermal, heat-sensing, far-infrared, and terahertz sensing applications. The study is overall compelling; therefore, I recommend its acceptance after the authors address the following points:

Answer: We sincerely thank Reviewer #1 for the positive and encouraging evaluation of our work, including the recognition that our study of an anomalously long thermal energy relaxation time in metallic $\text{Ti}_3\text{C}_2\text{T}_x$ is “a remarkable finding with significant scientific implications.” We also appreciate the reviewer’s supportive assessment that “the study is overall compelling” and the recommendation for acceptance pending revision. We address each point in detail below, with corresponding revisions implemented in the manuscript.

1.The manuscript should clearly distinguish between the mechanisms of “shielding” and “absorption” in the THz response. The claimed absorption of 80% significantly exceeds the theoretical limit (~50%) for intrinsically absorbing films with subwavelength thickness.

Answer: We thank the reviewer for pointing this out. We agree that we should have indeed stated “~80% extinction” which is referred to $1-T$ (where T is transmission) and includes both true absorption and reflection (shielding), rather than “~80% absorption”. To clarify this point, we have now explicitly separated transmittance (T), reflectance (R), and absorption (A) using the standard sheet-impedance model for a conductive thin film on a dielectric substrate (refractive index $n_s = 1.96$). [Nat. Photon. 2023, 17, 622–628] For a film much thinner than the THz wavelength (so multiple internal reflections within the film can be neglected), and with sheet resistance $R_s = \frac{1}{\sigma}$, the Fresnel coefficients at normal incidence are (air \rightarrow film \rightarrow substrate):

$$r = \frac{n_s - 1 - Z_0/R_s}{n_s + 1 + Z_0/R_s}, \quad t = \frac{2}{1 + n_s + Z_0/R_s}, \quad R = |r|^2, \quad T = n_s |t|^2, \quad A = 1 - R - T$$

where Z_0 is the wave impedance in vacuum, σ is the complex conductivity, n_0 is the refractive index of air, n_s is the refractive index of the substrate, and d is the thickness of the sample. This yields the total reflectance and transmittance of the sample-substrate system. The inferred

reflection and absorption are shown in **Fig. R1a**, where the red, blue, and green curves correspond to the reflectance R , transmittance T , and absorption A , respectively. Using the impedance-matching thin-film model [Nat. Photon. 2023, 17, 622–628)] with our experimentally determined average conductivity (1.6×10^6 S/m) and film thickness (~ 25 nm), we obtain a theoretical absorptance of $\sim 18.5\%$, which is in agreement with the experimentally measured value.

Figure R1. THz optical response of a ~ 25 nm $\text{Ti}_3\text{C}_2\text{T}_x$ film on a quartz substrate. The reflectance R (black), transmittance T (blue), and absorption A (red) shown here are calculated using intensity (power) values. Under this correct intensity representation, the film exhibits $\sim 79.1\%$ reflectance, $\sim 2.4\%$ transmittance, and $\sim 18.5\%$ true absorption across 0.5–2.2 THz.

Action: We have revised the manuscript to clearly differentiate extinction from true absorption. The original sentence: “The thin MXene films absorb more than 80% of the THz field.” has been replaced with “The thin $\text{Ti}_3\text{C}_2\text{T}_x$ films exhibit $\sim 18.5\%$ absorption across 0.5 to 2.2 THz, consistent with the thin-film electromagnetic limit (see **Fig. S1**).³⁰”

In addition, we have added the calculated reflectance (R), transmittance (T), and absorptance (A) spectra (now included as Figure S1) and described the full calculation procedure in the Supplementary Information.

2. The effective carrier mass value ($m^*=0.28 m_0$) used in the analysis is derived from theoretical calculations for $\text{Ti}_3\text{C}_2(\text{OH})_2$. However, experimentally synthesized $\text{Ti}_3\text{C}_2\text{T}_x$ typically contains a

mixture of various functional groups (e.g., -O, -OH, -F, -Cl). It is recommended that the authors perform Hall effect measurements to directly determine the carrier concentration, thereby enabling obtaining of the effective mass, and mobility by THz data, offering more reliable support for the charge transport properties.

Answer: We thank the reviewer for this insightful comment. We agree that the experimentally synthesized $\text{Ti}_3\text{C}_2\text{T}_x$ MXene contains mixed surface terminations (-O, -OH, -F, -Cl), which can influence the effective carrier mass (m^*). In the current manuscript, we used the DFT-calculated value of $m^* = 0.28 m_0$ for $\text{Ti}_3\text{C}_2(\text{OH})_2$ as a representative reference. Following the reviewer's suggestion, we have now clarified this explicitly in the revised version and discussed how variations in surface terminations could lead to slight modifications of m^* .

Furthermore, we emphasize that our main conclusions are based on experimentally measured intrinsic and photo-induced conductivities ($\Delta\sigma$), as well as their one-to-one correspondence between thermally and optically induced responses. These results do not rely on the exact numerical value of m^* , and therefore the interpretation of the long-lived photoconductivity remains robust.

Although we did not perform Hall effect measurements on the exact films reported in this study, prior Hall data on $\text{Ti}_3\text{C}_2\text{T}_x$ MXene (A. Miranda et al., Appl. Phys. Lett. 108, 033102 (2016)) involving some of us reported carrier densities of the same order of magnitude ($\sim 10^{21} \text{ cm}^{-3}$), consistent with our estimates.

Action: Following the reviewer's comment, we have added a clarification in the Methods section "The effective carrier mass $m^* = 0.28 m_0$ is used, which obtained from DFT-calculated value for $\text{Ti}_3\text{C}_2(\text{OH})_2$.⁴⁹ We note that experimentally synthesized $\text{Ti}_3\text{C}_2\text{T}_x$ contains mixed surface terminations (-O, -OH, -F, -Cl), which may alter the band dispersion and therefore m^* ."

3. The discussion regarding the underlying mechanism for the unusually slow thermal energy dissipation in $\text{Ti}_3\text{C}_2\text{T}_x$ is currently insufficient.

Answer: We thank the reviewer for raising this important point. The primary aim of our work is to establish a unified understanding of how photothermal effect influences non-equilibrium charge transport in $\text{Ti}_3\text{C}_2\text{T}_x$. While a complete quantitative decomposition of thermal energy dissipation pathways is an important question, it naturally requires dedicated thermal metrology and

systematic sample series, and therefore goes beyond the scope of the present study. Nevertheless, we agree that a clearer discussion of the likely thermal relaxation channels will strengthen the manuscript.

Given the thin nature of our multilayered MXene (with thicknesses of 25 nm and lateral sizes of several μm), the slow energy dissipation indicates that the thermal transport, at least in the out-of-plane direction is inefficient. This may originate from (i) a small interfacial (Kapitza) conductance G at the $\text{Ti}_3\text{C}_2\text{T}_x$ /substrate interface (*Nano Lett.* 2023, 23, 7, 2677–2686); and/or (ii) a low inter-layer vertical thermal conductivity κ_{\perp} between flakes (*Nano Lett.* 2024, 24, 51, 16333–16341). A rigorous separation of these two relaxation channels requires dedicated thermal metrology (e.g., TDTR/FDTR), systematic of sample thickness, and interface engineering, which we are pursuing as follow-up work.

Action: Following the reviewer’s comment, we have expanded the Discussion to include the following text:

“The relatively slow heat dissipation observed from near time zero to 100s of ns indicates that the dominant energy carrier is phononic heat, as electronic heat would dissipate much more rapidly. This supports the OOTP findings, which also directly point to a phononic character of the thermal energy. Given the thin nature of our multilayered MXene (with thicknesses of 25 nm and lateral size of several μm), the >100 ns slow heat dissipation implies a thermal transport bottleneck in our samples, at least in the thickness direction. This effect may originate from (i) a small interfacial (Kapitza) conductance across $\text{Ti}_3\text{C}_2\text{T}_x$ /substrate interfaces [*Nano Lett.* 2023, 23, 7, 2677–2686], and/or (ii) a low inter-layer thermal conductivity between flakes.[*Nano Lett.* 2024, 24, 51, 16333–16341] A quantitative separation of interfacial and cross-plane thermal conductance will require dedicated thermal metrology (e.g., TDTR/FDTR) studies with systematic thickness/substrate series, which we plan to address in a follow-up study.”

4. The manuscript lacks a direct comparison between the measured thermal energy relaxation time and those typical of conventional metals.

Answer: We appreciate this suggestion and agree a clearer benchmark is useful. To ensure a fair comparison, this discussion is confined to results obtained from OPTP measurements. Studies on the photoconductivity of metal thin films are scarce, likely because THz transmission through them is low unless their thickness is below ~ 10 nm. For instance, recent studies have reported negative photoconductivity dynamics with distinct timescales for different metals. In Cu and Cu-Pd alloys (with Pd < 50%), the response is extremely short-lived, decaying within approximately 2 ps.[J. Chem. Phys. 157, 174702 (2022)] In contrast, the induced negative photoconductivity in Au and Ti thin films persists for much longer, lasting from tens to a few hundreds of picoseconds. This long-lived component in Au nanofilms has been attributed to relatively slow interfacial heat flow.[Appl. Phys. Lett. 108, (2016).]

By contrast, in our ~ 25 nm $\text{Ti}_3\text{C}_2\text{T}_x$ films we observe lattice heating persisting beyond 100 ns, which is several order of magnitude longer than what is commonly reported for metal thin films of comparable or smaller thickness. This underscores that $\text{Ti}_3\text{C}_2\text{T}_x$ exhibits a markedly stronger thermal bottleneck, consistent with an interface-limited regime (small interfacial conductance) and/or low cross-plane thermal conductivity of stacked, terminated flakes. We will make this contrast explicit and cite the, Cu, Cu-Pd alloy, Ti and Au nanofilm study noted above.

Action: We have added a comparative paragraph to the Discussion:

“In conventional metals, ultrafast optical studies typically reveal a multistep relaxation: sub-ps electron–electron thermalization, followed by sub-ps to ps electron–phonon energy transfer, and finally ps-to-hundreds-of-ps lattice cooling. While this cooling in ultrathin films ($\leq \sim 10$ nm) is often reported to be shortly lived from few to 100s of ps depending on the metal nature and its interfaces^{12,15}, our ~ 25 nm $\text{Ti}_3\text{C}_2\text{T}_x$ films exhibit a drastically slow dynamics persisting for >100 ns. This orders-of-magnitude difference indicates a severe thermal bottleneck, which may be attributed to low interfacial conductance and/or low in-plane thermal conductivity in our MXenes^{35,37,45} and a very low emissivity of $\text{Ti}_3\text{C}_2\text{T}_x$ ⁴⁶.”

Reviewer #2 (Remarks to the Author):

This manuscript addresses the question of the origin of the anomalously long relaxation lifetime in Ti_3C_2 MXene films. It concludes that the negative photoconductivity arises from thermal effects, consistent with previous studies using time-resolved diffraction and transient absorption. The novelty and significance of this work lie in its comprehensive dataset, which presents photoconductivity measurements across an exceptionally broad range of photon energies, from UV to IR. This contribution is both timely and relevant, offering valuable insights into MXene properties. The manuscript could be further strengthened by expanding the discussion on several points of interest to the community, which would not require additional experiments.

Answer: We sincerely thank Reviewer #2 for the thoughtful and constructive evaluation of our work. We are especially grateful for the recognition that the study is “*both timely and relevant,*” and that it “*offers valuable insights into MXene properties.*” We appreciate the reviewer’s insightful comments regarding the interpretation of our excitation-energy–dependent data, and the suggestion to expand the discussion to further contextualize the observed dynamics. We have carefully considered all points raised and have revised the manuscript accordingly. Detailed, point-by-point responses are provided below, together with the specific changes implemented in the revised text.

1. In the THz TDS measurements, the conductivity exhibits a Drude-like response, without the low-frequency suppression or negative imaginary conductivity reported in prior studies which used Drude-Smith model to analyze the conductivity. This result suggests a significant improvement in film quality, where flake edges or other defects no longer hinder long-range transport. It would be valuable if the authors could comment on which advances in MXene synthesis or film deposition they attribute to this improvement. A more detailed methods section outlining the steps necessary to achieve such high AC conductivity would also be highly appreciated by the community.

Answer: We thank the reviewer and agree that the purely Drude response indicates improved long-range transport with minimal back-scattering at e.g., interflake junctions/edges. $\text{Ti}_3\text{C}_2\text{T}_x$ MXene is a well-established 2D material system, and the synthesis protocol used in this work follows the widely adopted HF/HCl etching of Ti_3AlC_2 MAX phase followed by LiCl-assisted delamination,

as reported in ACS Nano 2021, 15, 6420–6429. This procedure reliably yields high-quality, few-layer $\text{Ti}_3\text{C}_2\text{T}_x$ flakes with minimal residual MAX phase, and is routinely used in our laboratory with excellent reproducibility.

In our work, the improved AC transport behaviour primarily arises from (1) the synthesis procedure and (2) the film assembly conditions:

(1) Synthesis and Delamination.

The $\text{Ti}_3\text{C}_2\text{T}_x$ MXene flakes used here were prepared following the well-established HF/HCl etching and LiCl-assisted delamination protocol reported in ACS Nano 2021, 15, 6420–6429. This method is known to yield highly delaminated, few-layer $\text{Ti}_3\text{C}_2\text{T}_x$ flakes with minimal residual MAX phase. The samples used in this work originate from the same synthesis batch as those reported in Adv. Electron. Mater. 2025, 2500017, ensuring reproducibility.

(2) Film Fabrication.

The thin films were prepared by solution spraying under controlled drying conditions, which promotes dense flake packing and strong interflake coupling, reducing junction resistance.

To directly address the reviewer’s comment, we have now included representative structural and morphological characterization data for $\text{Ti}_3\text{C}_2\text{T}_x$ thin films prepared using the same synthesis method and processing conditions. These data confirm the expected crystallographic features, flake morphology, and surface state of high-quality $\text{Ti}_3\text{C}_2\text{T}_x$ MXene films:

Figure R2. (a) X-ray diffraction (XRD) pattern of $\text{Ti}_3\text{C}_2\text{T}_x$, showing a dominant (002) reflection characteristic of well-delaminated MXene and confirming successful etching of the Ti_3AlC_2 MAX precursor. (b) Raman spectrum of a $\text{Ti}_3\text{C}_2\text{T}_x$ film obtained using a 785 nm excitation laser, displaying the expected in-plane and out-of-plane vibrational modes associated with Ti–C bonding and surface terminations. (c) Transmission electron microscopy (TEM) image of a freshly prepared $\text{Ti}_3\text{C}_2\text{T}_x$ flake, showing clean and well-layered morphology. Panels (a–c) are adapted with permission from Ref. [Adv.

In **Fig. R2**, XRD measurements of the films show a dominant (002) reflection with the expected interlayer spacing for $\text{Ti}_3\text{C}_2\text{T}_x$, and no detectable TiO_2 -related secondary phases. Raman spectroscopy displays the characteristic vibrational modes associated with Ti–C bonding and surface terminations in $\text{Ti}_3\text{C}_2\text{T}_x$, consistent with well-preserved MXene structure. [Chem. Mater. 2023, 35, 8239.] Furthermore, TEM images further reveal clean, well-layered flakes without obvious structural degradation or particulate oxidation. Together, these characterizations confirm that the $\text{Ti}_3\text{C}_2\text{T}_x$ films used in this work possess the crystallographic integrity and flake quality expected for reliable optical and transport measurements.

Importantly, the samples used in the present study originate from the same synthesis batch as those characterized in our previous work [Adv. Electron. Mater. 2025, 2500017] and were processed using the same dispersion and film fabrication procedures.[ACS Nano 2021, 15, 6420–6429] Therefore, the structural and morphological data shown in Figure R2 are directly representative of the films measured here. We have added explicit cross-references to these characterizations in the main text to clarify this connection.

2. The authors further observe that carrier scattering rates decrease substantially as the temperature is lowered, accompanied by a reduction in conductivity. In contrast, other studies have reported that Ti_3C_2 conductivity can increase again at low temperatures (50–100 K) after an initial decrease (see, for example, Anasori et al, Nanoscale Horiz. 2016, 1, 227; Figure 3B). Here, the data clearly show no such reversal. Does this difference again highlight film quality as the determining factor? The authors' interpretation would greatly aid the community in deepening the understanding of conductivity mechanisms in these materials.

Answer: Thank you for this thoughtful point. The low-temperature “upturn” suggested by the reviewer was reported from dc/device measurements, whereas our data are obtained by high-frequency THz time-domain spectroscopy. These probes emphasize different transport length

scales and bottlenecks: THz (0.3–2 THz) predominantly senses short-range, intra-flake conduction over tens of nanometers, while *dc* transport reports charge carrier percolation across inter-flake junctions/edges and macroscopic connectivity. Consequently, different, and sometimes even opposite trends can arise without contradiction. For example, our recent reports unveil seemingly "contradictory" transport mechanisms in the same sample, including MXene when comparing THz and *dc* transport data (band-like vs hopping). [Nature Physics, 2022, 18(5): 544-550.; Advanced Materials, 2023, 35(15): 2211157; ACS nano, 2022, 16(6): 9401-9409.] In our films the THz response reflects phonon-limited intra-flake mobility and does not show a low-*T* reversal; device-level *dc* conductance can exhibit additional behaviour set by inter-flake hopping, junction resistance, intercalants/solvents, and oxidation states. We will make this distinction explicit and cite prior work establishing that THz probes intra-flake transport whereas *dc* is limited by inter-flake pathways.

Regarding the reviewer's question, we note that film quality is unlikely to be the determining factor for this difference. While high-quality flakes help ensure intrinsic behaviour, the dominant reason for the contrasting trends lies in the different physical quantities probed by *ac* (THz) and *dc* techniques—intra-flake vs. inter-flake transport.

3. In the OPTP section, the authors present compelling data on the excitation energy dependence of NPC conductivity dynamics. They show that the sub-picosecond component becomes more pronounced at higher excitation photon energies. This dataset substantially extends the previously reported comparison of 400 nm versus 800 nm excitation in Ref. [25]. A question to the authors: in Ref. [25], the rise time of NPC also differed between the two wavelengths. Do you observe a similar effect here? From Fig. 3a, it is difficult to determine. If such an effect is present, is there a clear transition from slower to faster rise times, or is the change monotonic with excitation energy? Any comments on the possible mechanisms driving this behavior would be highly valuable.

Answer: In our data, the rise times for 800 nm (1.55 eV) and 400 nm (3.10 eV) excitation are indistinguishable within our temporal resolution (≈ 200 fs). This is evident in **Fig. R3a**, where both transient photoconductivity traces overlap closely during the sub-picosecond onset.

This observation differs slightly from Ref. [25], where a wavelength-dependent rise time was reported. We note, however, that in Ref. [25] the difference between 400 nm and 800 nm

excitations becomes noticeable only at high pump fluences (\geq hundreds $\mu\text{J cm}^{-2}$), where nonlinear heating and multi-photon effects may influence the early-time carrier dynamics. In our experiments, by contrast, the absorbed photon density (N_{abs}) was carefully adjusted across all excitation wavelengths to remain sufficiently low to be within the linear regime, which allows us to study the intrinsic role of pump photon energy in dictating carrier dynamics in MXenes.

Figure R3. (a) OPTP photoconductivity dynamics of $\text{Ti}_3\text{C}_2\text{T}_x$ MXene films excited at 3.10 eV (400 nm) and 1.55 eV (800 nm) normalized by the absorbed photon density (N_{abs}). The initial rise time and sub-picosecond dynamics are essentially identical for the two excitation energies, indicating that the early-stage carrier–phonon equilibration is not strongly dependent on photon energy under low-fluence conditions. (b) Transient transmission ($\Delta T/T$) dynamics of $\text{Ti}_3\text{C}_2\text{T}_x$ films reported in Ref. 25 (*Nano Lett.* 2020, 20, 636–643). At low excitation fluence, the rise times at 400 nm and 800 nm are likewise similar, consistent with our observation. Differences reported at high fluence in Ref. 25 arise from fluence-dependent heating effects rather than intrinsic wavelength dependence. The vertical dashed line is provided as a visual guide to compare the rise dynamics. Panel (b) is adapted with permission from Ref. 25 (*Nano Lett.* 2020, 20, 636–643). © 2020 American Chemical Society.

4. The central findings and novelty of this work largely stem from the excitation energy–dependent measurements presented in Figure 3. The optical absorption of Ti_3C_2 varies significantly across the

0.5–4.5 eV range explored here, from interband transitions in the UV, to free-carrier absorption in the IR, to the pronounced peak near 750–800 nm attributed to plasmon resonance, which the authors use to extract thermal dynamics from transient reflectivity. A key question is whether the authors accounted for this variation in optical absorption when estimating the initial deposited energy via photoexcitation (N_{abs}). The energy-dependent relaxation reported in Fig. 3b appears linear and does not correlate with the expected absorption spectrum. Why is that? Are there differences in the earlier dynamics (sub-2 ps) that are not captured in the presented analysis? Intuitively, one might expect that exciting interband transitions versus resonant plasmon excitation would yield distinct relaxation pathways at early times. Since the authors already have the relevant data, a discussion of this point, ideally supported by a UV–Vis spectrum, would provide valuable additional insight.

Answer: We thank the reviewer for this detailed and thoughtful question.

(i) Accounting for absorption when defining N_{abs} . Yes, we have fully accounted for the wavelength dependence of the optical absorption when determining the absorbed photon density (N_{abs}). The absorbed photon number was calculated as $N_{abs} = (F \times A) / (h\nu)$, where F is the incident fluence, $A(\lambda)$ is the measured absorption at the corresponding wavelength, and $h\nu$ is the photon energy. The fluence was adjusted at each excitation wavelength to ensure that N_{abs} remained similar. This normalization allows us to directly compare energy relaxation dynamics across photon energies without bias from spectral variations in absorption.

(ii) Why the amplitude after 2 ps relaxation in Fig. 3b appears linear and does not follow the absorption spectrum. The apparent linear dependence arises precisely because the absorption variation has been normalized out by fixing N_{abs} . In other words, since all datasets correspond to the same number of absorbed photons, the plateau amplitudes following fast initial decay scale linearly with the photon-energy $h\nu$ rather than the raw optical absorption spectrum.

(iii) Early-time (sub-2 ps) differences.

Yes, differences are present in the ultrafast regime (sub-2 ps), as shown in Figure S3. We find that higher photon energies (e.g., 3.1 eV) yield significantly greater spectral weight in the fast dynamics. These early-time dynamics decay within approximately 2–3 ps, after which the signal stabilizes and its amplitude remains constant for a given pump fluence ($N_{abs} \cdot h\nu$). We therefore speculate

that this picosecond-scale dynamics likely reflects the relaxation between hot carriers and the lattice. This assignment is also in line with recent ultrafast studies by transient absorption or X-ray diffraction.[Nat Commun. 13, 7900, (2022); Nano Lett. 23, 2677-2686, (2023).] By the end of this process, the carriers and lattice reach a thermal equilibrium, resulting in the lattice heating effect discussed on the millisecond timescale.

(iv) Interband versus plasmonic excitation.

We agree that interband excitation (in the UV–visible) and plasmonic excitation (near 700–850 nm) can initially follow distinct relaxation pathways.[Nat Commun. 13, 7900, (2022).] However, both channels rapidly funnel energy into the same phonon bath through carrier–phonon coupling and nonradiative plasmon decay, respectively. Once this equilibration is complete, the subsequent photothermal dynamics are governed solely by the total energy deposited into the lattice, independent of the initial optical excitation pathway. This inherent independence explains the consistent, long-lived photothermal response observed across different photon energies.

Action: We have added a clarifying paragraph to the Method section:

“To make a fair comparison for the photoconductivity, we normalized photoconductivity dynamics to the absorbed photon density N_{abs} . The wavelength-dependent absorption is obtained by:

$$N_{abs}(h\nu) = \frac{P_{avg}/f}{h\nu} \times A(h\nu) \div A_{spot}$$

where P_{avg} is the average laser power, f the laser repetition rate (500 Hz) after optical chopper, and A_{spot} the pump spot area overlapped with our THz probe, which we determined by using a 1 mm diameter calibrated iris aperture, $A(h\nu)$ is the film absorption measured by UV–Vis–NIR spectrophotometry with an integrating sphere. Before this, we checked the fluence dependence at each wavelength and selected a fluence range where the OOTP signal is linear.”

And added the corresponding discussion in the Results section:

“We attribute this fast initial decay to the ultrafast energy relaxation between thermalized/nonthermalized hot carriers to the lattice, resulting in ultrafast heating of the lattice, which is in line with recent ultrafast studies by TA and X-ray scattering studies.^{34,36,39} Although different excitation schemes (e.g., interband vs. plasmonic) may initiate distinct energy relaxation

pathways, all the energy ultimately deposits into a common phonon bath within this 2–3 ps window (corresponding to the process from stage I to II in **Fig. 1**). The later conductivity recovery (stage II→III) occurs on timescales exceeding 100 ns, corresponding to slow lattice heat dissipation.”

Small comment:

In the legend of Fig. 3, please specify the excitation energy for panel C (and vertical cut in Panel e)

Answer: We thank the reviewer for the helpful suggestion. We have now added the excitation photon energy and temperature to the legends of Fig. 3 panels a–e.

Reviewer #3 (Remarks to the Author):

Zheng et al. conducted a systematic investigation of the photoexcitation dynamics in $\text{Ti}_3\text{C}_2\text{T}_x$ MXenes, employing a combination of optical pump–terahertz probe (OPTP) spectroscopy and transient reflectance (TR) measurements. Their study revealed that photothermal effects play a dominant role in the unusually long-lived negative photoconductivity (NPC) observed in MXene. The main innovation of the work lies in proposing a unified framework that connects photothermal conversion, electron–phonon interactions, and thermal relaxation with charge transport processes, supported by comprehensive experimental data. This work provided significant insights into the photophysical mechanisms of two-dimensional metallic MXenes and offered valuable guidance for their potential optoelectronic and thermoelectric applications. The analysis is mostly convincing, and thus the work is in principle well suited for publication in Nature Communication. Nevertheless, the manuscript may have several weaknesses and requires additional description and analysis to strengthen its conclusions.

Answer: We sincerely thank Reviewer #3 for the thoughtful and constructive evaluation of our work. We appreciate the recognition of our unified framework linking photothermal conversion, electron–phonon coupling, and thermal relaxation to charge transport dynamics, as well as the comment that the study provides meaningful insights for potential optoelectronic and thermoelectric applications of MXenes. We are grateful for the reviewer’s positive assessment that

the analysis is convincing and well suited for publication in *Nature Communications*. We address the specific points raised below and have revised the manuscript accordingly to strengthen clarity and completeness.

Q1. The current version of the manuscript lacks sufficient structural and morphological characterization of the $\text{Ti}_3\text{C}_2\text{T}_x$ MXene samples. For example, techniques such as X-ray diffraction (XRD) or transmission electron microscopy (TEM) are needed to verify the sample quality and crystallinity. Without these data, it may be difficult to fully assess the reliability and reproducibility of the observed photoexcitation dynamics.

Answer: We appreciate the reviewer's request for clearer evidence of sample quality and reproducibility. As noted in our response to Reviewer #2, the $\text{Ti}_3\text{C}_2\text{T}_x$ flakes used here were synthesized and delaminated using a well-established HF/HCl etching followed by LiCl-assisted delamination protocol [ACS Nano 2021, 15, 6420–6429]. The spray-coated ~25 nm films on quartz measured in this work were prepared from the same synthesis batch and using the same deposition/conditioning procedure as those structurally characterized in Adv. Electron. Mater. 2025, 2500017. For completeness, Figure R2 (cited in our response to Reviewer #2) compiles representative XRD, Raman, and TEM from that batch: XRD shows the expected layered MXene signature with a strong (002) reflection; Raman displays the characteristic Ti–C/termination modes reported for $\text{Ti}_3\text{C}_2\text{T}_x$; and TEM reveals clean, well-layered flakes consistent with high-quality delaminated MXene. Because the films analysed here and those characterized previously are from the same batch and processed identically, the structural/morphological data in Figure R2 are directly representative of the samples used for our optical measurements. We have added explicit cross-references in the Methods to make this linkage clear.

Q2. As illustrated in Fig. 1, the authors suggested that increasing the temperature reduces the mean free path of charge carriers, which is a reasonable description. In this study, based on the available data, is it possible to estimate the mean free path values of MXene charge carriers in the three stages shown in Fig. 1?

Answer: We thank the reviewer for this thoughtful question. The mean free path (l) of charge carriers can be estimated from our temperature-dependent Drude parameters obtained via THz-TDS. Within the Drude model framework,

$$l(T) = v_F \tau(T), \text{ where } v_F = \frac{\hbar k_F}{m^*}, k_F = (3\pi^2 n)^{1/3}$$

Here:

- \hbar is the reduced Planck constant,
- k_F is the Fermi wavevector,
- n is the carrier concentration,
- m^* is the effective carrier mass,
- v_F is the Fermi velocity,
- $\tau(T)$ is the temperature-dependent carrier scattering time, and
- $l(T)$ is the transport mean free path.

Using $m^* = 0.28 m_0$ and $n \approx 1.2 \times 10^{21} \text{ cm}^{-3}$, we obtain $v_F \approx 1.56 \times 10^6 \text{ m/s}$. From the THz fits, the scattering time $\tau(T)$ varies from $\sim 10 \text{ fs}$ at 300 K to $\sim 15 \text{ fs}$ at 100 K, leading to estimated in-plane mean free paths of $\sim 14 \text{ nm}$ at 300 K and $\sim 23 \text{ nm}$ at 100 K. Thus, heating (either optically or thermally) effectively shortens the mean free path by increasing phonon-mediated scattering.

In the context of Fig. 1:

- Stage I \rightarrow II (photoexcitation and rapid lattice heating) corresponds to a transient reduction of mean free path from $l(T)$ (between 14 to 23 nm depending on T) to $l(T + \Delta T)$;
- Stage II \rightarrow III (thermal relaxation) marks a gradual recovery as the system cools, restoring longer mean free paths.

While these values are approximate, they nevertheless provide a meaningful order-of-magnitude indication of the transport regime probed by THz spectroscopy, reinforcing our conclusion that photothermal excitation chiefly alters scattering rates rather than the carrier density.

Action: We have added a short quantitative paragraph to the Results section:

“This behaviour suggests the central role of electron–phonon interactions in governing the electrical response of the system. Using the scattering times obtained from the Drude analysis, we estimate the in-plane mean free path as $l(T) = v_F \tau(T)$, where $v_F = \frac{\hbar k_F}{m}$ and $k_F = (3\pi^2 n)^{1/3}$. Here \hbar , k_F , v_F are the reduced Planck constant, Fermi wavevector and Fermi velocity. Based on these parameters, the mean free path ranges from ~ 14 nm at 300 K to ~ 23 nm at 100 K.”

Q3. In the manuscript, the authors state: “Our ultrafast conductivity studies pinpoint the efficient photothermal response (Fig. 1, I to II) and slow thermal energy dissipation (Fig. 1, II to III) as the main origins of the observed long-lived NPC effect.” Could more accurate time scales for these two processes be extracted through quantitative fitting of the dynamical curves?

Answer: We appreciate the reviewer’s request for clarification regarding the characteristic timescales. Based on the photon-energy–dependent OPTP dynamics, we can assign physically meaningful time windows to the two processes depicted in Fig. 1:

- Stage I \rightarrow II (carrier–phonon thermalization): The rise and early decay of the photoconductivity occur within ~ 3 ps, reflecting the rapid transfer of energy from hot carriers to the lattice. This timescale is consistent across excitation photon energies and agrees with ultrafast carrier–phonon equilibration reported in metallic MXenes. [Nano Lett. 24, 16333-16341, (2024).; Nano Lett. 23, 2677-2686, (2023).; ACS Nano 18, 32491-32497, (2024).; Adv. Mater., 2208659, (2022)]
- Stage II \rightarrow III (lattice heat dissipation): The slow recovery of conductivity occurs on much longer timescales, extending from tens of picoseconds to >100 ns, as determined from the repetition-rate–dependent transient reflectance measurement (Fig. 4). This regime reflects the gradual dissipation of lattice heat into the substrate and environment.

We note that extracting these two timescales via direct multi-exponential curve fitting would not be rigorous: The fast decay involves complex hot carrier-lattice dynamics with time-evolving carrier mobility, making it challenging to fit the dynamics. Conversely, the slow thermal dynamics

occur on a timescale that exceeds the measurement window of our THz studies, preventing us from determining its precise duration. Instead, the time scale of slow thermal relaxation was done by transient reflection spectroscopic measurements. Therefore, we would prefer not to fit the current THz dynamics, and only report timescales based on the physically identifiable temporal regimes.

Action: We have added the following clarification to the Results and Discussion section:

“Although different excitation schemes (e.g., interband vs. plasmonic) may initiate distinct energy relaxation pathways, all the energy ultimately deposits into a common phonon bath within this 2–3 ps window (corresponding to the process from stage I to II in **Fig. 1**). The later conductivity recovery (stage II→III) occurs on timescales exceeding 100 ns, corresponding to slow lattice heat dissipation.”

Q4. The authors state: Fig. 3a illustrates the $h\nu$ -dependent photoconductivity dynamics in Ti3C2Tx MXene thin films with the same absorbed photon density (N_{abs}). It would be helpful if the paper could provide a clear definition of absorbed photon density as well as the calculation method used to ensure the same absorbed photon density under different excitation wavelengths. Since MXenes exhibit varying absorption strengths across different spectral regions, it is important to explain how the authors corrected for these differences to guarantee consistent absorbed photon density across all wavelengths.

Answer: Thank you for this insightful question. We did account for the wavelength-dependent absorption when comparing Fig. 3. For each pump wavelength we calculated the absorbed photon density as

$$N_{abs}(h\nu) = \frac{P_{avg}/f}{h\nu} \times A(h\nu) \div A_{spot}$$

where P_{avg} is the average laser power, f the laser repetition rate (500 Hz) after optical chopper, and A_{spot} the pump spot area overlapped with our THz probe, which we determined by using a 1 mm diameter calibrated iris aperture, $A(h\nu)$ is the film absorption measured by UV–Vis–NIR spectrophotometry with an integrating sphere. Before this, we checked the fluence dependence at each wavelength and selected a fluence range where the OOTP signal is linear.

The photoconductivity change $\Delta\sigma(h\nu, t)$ presented in Fig. 3 is then normalized as:

$$\Delta\sigma_{norm}(h\nu, t) = \frac{\Delta\sigma(h\nu, t)}{N_{abs}(h\nu)}$$

which allows meaningful comparison across excitation photon energies despite differences in absorption. Thus, the apparent linear trend in Fig. 3b reflects equivalent absorbed photon densities after normalization, rather than the raw absorption spectrum.

Action: In response to the reviewer's suggestion, we have added a clear definition of N_{abs} and a detailed description of the normalization procedure to the Methods section:

“To make a fair comparison for the photoconductivity, we normalized photoconductivity dynamics to the absorbed photon density N_{abs} . The wavelength-dependent absorption is obtained by:

$$N_{abs}(h\nu) = \frac{P_{avg}/f}{h\nu} \times A(h\nu) \div A_{spot}$$

where P_{avg} is the average laser power, f the laser repetition rate (500 Hz) after optical chopper, and A_{spot} the pump spot area overlapped with our THz probe, which we determined by using a 1 mm diameter calibrated iris aperture, $A(h\nu)$ is the film absorption measured by UV-Vis-NIR spectrophotometry with an integrating sphere. Before this, we checked the fluence dependence at each wavelength and selected a fluence range where the OTP signal is linear.”